# FedRAC: Rolling Submodel Allocation for Collaborative Fairness in Federated Learning

## Abstract

Collaborative fairness in federated learning ensures that clients are rewarded according to their contributions, thereby fostering long-term participation among clients. However, existing methods often under-reward low-contributing clients in the early training stage and neglect critical issues, such as consistency across local models (i.e., inter-model inconsistency) or unequal neuron training frequencies in the aggregated model (i.e., intra-model inconsistency), both of which lead to degraded performance. To address these issues, we propose FedRAC, a novel Federated learning framework employing Rolling submodel Allocation for Collaborative fairness, without compromising the global model performance. First, we design a *dynamic reputation calculation module* with a theoretical fairness guarantee to generate reputations matching clients' contributions. It adjusts their reputations dynamically during training, ensuring low-contributing clients access better models in the early stages for adequate training. Second, we propose a *rolling submodel allocation module* that assigns high-performance submodels to clients with high reputations. This module prioritizes low-frequency neurons during allocation and is supported by theoretical convergence guarantees, ensuring that all neurons in the global model are fully trained. Extensive experiments are conducted on three public datasets to confirm the advantages of our method in terms of fairness and model accuracy.

## 1 Introduction

Federated Learning (FL) has emerged as an effective distributed machine learning framework that enables multiple clients to train a global model collaboratively while preserving their privacy McMahan et al. (2017); Li et al. (2020); Wang et al. (2024b; 2025). However, in real-world scenarios, clients contribute unequally to the system as their data varies in scale and quality Wang et al. (2024c); Liu et al. (2022); Fu et al. (2023). Early FL frameworks typically provided all clients with the same models, overlooking their contributions and discouraging high-contributing clients Guan et al. (2024); Yu et al. (2025); Yan et al. (2024). Collaborative fairness (CF) serves as a critical component of FL, motivating client engagement by ensuring that the allocated global model aligns with individual contributions Wang et al. (2024e); Lyu et al. (2020).

Existing collaborative fairness methods, which generally involve reputation calculation and reward allocation, can be divided into two groups. The first group (i.e., gradient-based) assigns clients fixed reputations according to their contributions during training. Then, clients are rewarded with a proportion of the aggregated gradients based on their reputations Lyu et al. (2020); Xu et al. (2021); Wang et al. (2024d); Wu et al. (2024b). Meanwhile, the second group (i.e., submodel-based) follows a similar reputation-based mechanism, but allocates submodels containing important neurons to clients in proportion to their reputations Wang et al. (2024e). However, despite significant progress in achieving CF, these methods still face two major limitations in attaining high model accuracy:

**In the reputation calculation phase**, existing approaches overlook the fact that the performance of the global model improves progressively. By assigning fixed reputation proportions to all clients throughout training, they under-reward low-contributing clients in the early stages, which ultimately leads to a decline in the performance of the aggregation model. As shown in Figure 1(a), the perfor-

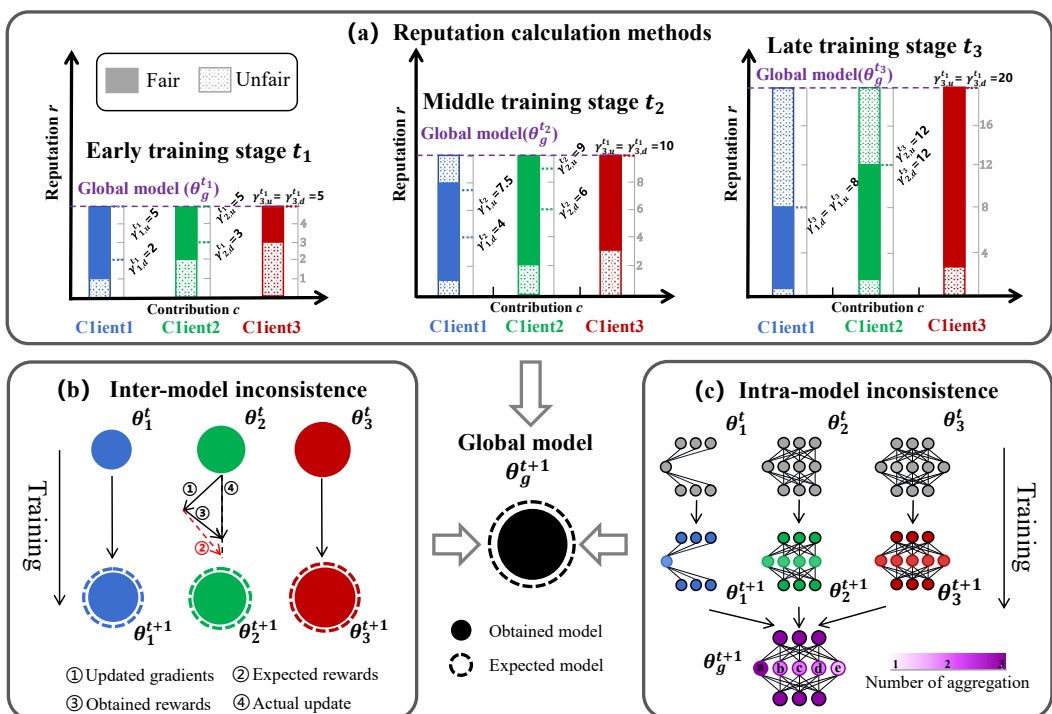

Figure 1: **Problem illustration of collaborative fairness in FL.** (a) Existing methods assign fixed reputation weights to clients throughout training, under-rewarding low-contributing clients in the early stages. (b) Gradient-based methods fail to maintain consistency across local models. (c) Submodel-based methods cause unequal training frequencies for neurons in the aggregated model. All three issues undermine the performance of the global model.

mance of the global model improves progressively ($\theta_g^{t_3} > \theta_g^{t_2} > \theta_g^{t_1}$), whereas the ratio of reputation scores across clients remains fixed ($r_{1,d}^{t_1} : r_{2,d}^{t_1} : r_{3,d}^{t_1} = r_{1,d}^{t_2} : r_{1,d}^{t_2} : r_{3,d}^{t_2} < r_{1,u}^{t_1} : r_{2,u}^{t_1} : r_{3,u}^{t_1}$). Consequently, $Client_1$ receives insufficient rewards in the early training stage $t_1$, which ultimately degrades the overall model performance. **In the reward allocation phase**, conventional methods often lead to inter-model (i.e., gradient-based) or intra-model (i.e., submodel-based) inconsistencies, thereby degrading the performance of the aggregated model. For example, in Figure 1(b), the gradient-based methods can cause substantial divergence among clients' models ($\theta_1^t$, $\theta_2^t$, and $\theta_3^t$). The gradients exchanged by the clients may not align with each other's needs, resulting in a mismatch between the obtained (i.e., ③) and expected(i.e., ②) rewards. Similarly, as shown in Figure 1(c), the submodel-based method can induce unequal training frequencies of neurons in the aggregated model, leaving some neurons (i.e., neuron $b$, $c$, and $e$) not being trained sufficiently.

To address these issues above, we propose **FedRAC**, *a novel Federated learning framework with Rolling submodel Allocation for Collaborative fairness*, designed to achieve competitive model accuracy. First, *the dynamic reputation calculation module*, theoretically guaranteed to ensure fairness, is designed to ensure that clients with higher contributions receive higher reputations. In particular, the reputations of clients are dynamic and gradually diverge during training, ensuring lower-contributing clients can still receive better rewards during the early stages of training. Second, *the rolling submodel allocation module* is proposed to allocate high-performance submodels to clients with higher reputations. This module prioritizes low-frequency (i.e., allocation count) neurons and is supported by theoretical convergence guarantees, ensuring that all neurons in the global model are effectively trained. Extensive experiments on three public benchmarks demonstrate that FedRAC surpasses all baseline methods in both collaborative fairness and model accuracy.

The contributions of this work are summarized as follow:

- We propose FedRAC, a novel federated learning framework that enhances collaborative fairness by fairly allocating appropriate rewards to clients while achieving competitive

model accuracy. To the best of our knowledge, this is the first approach to ensure an optimal balance between fairness and model accuracy.

- In the reputaion calculation phase, we design a dynamic reputation calculation module, under a theoretical fairness guarantee, that aligns clients' reputations with their contributions. The module dynamically updates reputations during training, enabling low-contributing clients to obtain better models in the early stages for sufficient training.

- In the reward allocation phase, we propose a rolling submodel allocation module, backed by a theoretical convergence guarantee, which allocates high-performance submodels to high-reputation clients. By prioritizing low-frequency neurons during allocation, this module ensures that all neurons of the global model are evenly trained across clients' local data.

- We conduct comprehensive experiments on three datasets (CIFAR-10, SVHN, and EM-NIST) to demonstrate FedRAC's superiority over state-of-the-art methods in both accuracy and fairness.

## 2 RELATED WORKS

**Model-heterogeneous.** In FL, clients are often resource-constrained edge devices Caldas et al. (2018); Liang et al. (2025). HeteroFL Diao et al. (2020) and FjORD Horvath et al. (2021) introduce two static submodel extraction methods to address network heterogeneity. In FL, the global model is restricted by the limited computational resources of client devices, and static submodel extraction hinders its complete training on clients. FedRolex Alam et al. (2022) proposes a rolling submodel extraction scheme. This method effectively addresses the limitations of both random and static extraction strategies. ScaleFL Ilhan et al. (2023) takes a different approach by dynamically adjusting the scale of the global model in each round. This accommodates the heterogeneous computing capabilities of clients. FedDSE Wang et al. (2024a) tackles the problem of clients being forced to adapt to new distributions under submodel allocation. It extracts submodels based on each client's data distribution, thereby reducing conflicts between clients. However, the above methods primarily focus on issues such as clients' computational and communication limitations. They do not consider the clients' contributions, which can lead to unfairness in the system.

**Collaborative fairness.** Collaborative fairness (CF) is a key research area in FL. It aims to reward clients based on their contributions, encouraging them to provide high-quality data or model updates. This mechanism ensures fairness and supports the long-term operation of the system. Client reputation is a crucial criterion for evaluating fairness in FL systems. CFFL Lyu et al. (2020) quantifies client reputation by the amount of data they contribute and assigns aggregation gradients proportionally, thereby achieving CF. CGSV Xu et al. (2021) introduces a reward calculation method based on gradient similarity and allocates rewards accordingly. FedAVE Wang et al. (2024d) addresses the limitation of existing methods that overlook clients' data distribution information. It proposes an FL framework that treats local data distribution as their reputations and dynamically allocates gradients accordingly. IAFL Wu et al. (2024b) mitigates the risk of high-reputation clients being contaminated by low-reputation clients during training. It ensures that high-contribution clients receive higher-quality model updates, while guaranteeing convergence of all clients to the global optimal solution. FedSAC Wang et al. (2024e) proposes a novel approach that assigns submodels as rewards to clients, thereby avoiding the reward conflicts arising from gradient allocation. However, these methods typically assume that client reputation remains static during training and cause inter/intra-model inconsistencies, both of which ultimately degrade the performance of the aggregated model. In contrast, we propose a novel framework for achieving CF by allocating appropriate rewards to clients while ensuring comprehensive model training. This framework can be effectively applied across diverse FL scenarios.

## 3 PRELIMINARY

### 3.1 FEDERATED LEARNING

In a FL system, there typically exists one server and multiple clients Yu et al. (2025); Wu et al. (2024a); Wang et al. (2021). These components collaboratively optimize a global model with the goal of minimizing the weighted average of all clients' local objective functions Yazdinejad et al.

(2024); Huang et al. (2024); Ye et al. (2024). It can be expressed as:

$$\min_{\theta} F(\theta) := \sum_{i=1}^{N} p_i F_i(\theta), \tag{1}$$

where $p_i$ represents the weight of each client $i$, $F_i(\theta)$ denotes the local loss function of client $i$ under the model $\theta$, and the total number of clients is $N$. Although FedAvg(McMahan et al. (2017)) has been proven effective in minimizing the global objective function, it overlooks disparities in clients' contributions Chen et al. (2024); Pan et al. (2024); Zhou et al. (2021); Sun et al. (2023); Fan et al. (2022); Li et al. (2024). By assigning identical rewards to all clients regardless of their contributions, this framework ultimately leads to unfairness in the FL system Ezzeldin et al. (2023); Zhang et al. (2025).

## 3.2 COLLABORATIVE FAIRNESS

Collaborative fairness in FL is a key consideration during model training, as it ensures high-contribution clients receive greater rewards to better incentivize their participation in collaboration. Specifically, inspired by FedSAC Wang et al. (2024e), we introduce the concept of $\alpha$-Bounded Collaborative Fairness ($\alpha$-BCF), which guarantees $c_i < \theta_i^* \leq [(1-\alpha)\frac{(c_i)}{\max(c)} + \alpha]\theta^*$. We further quantify fairness using the Pearson Correlation Coefficient $\rho(c, \theta^*)$, where $c_i$ denotes the contribution of the $i$-th client and $\alpha$ is a hyperparameter. The rationale for the formula in Definition 1 is to highlight meaningful differences in rewards. Specifically, the left side of the formula ensures each client's reward is higher than their contribution, while the right side stops clients with low contributions from obtaining excessive rewards. Meanwhile, when $\alpha = 0$, the right side of the formula ensures a client's reward can exceed their own contribution $c_i$; when $\alpha = 1$, the reward is capped below the ideal maximum reward $\theta^*$.

**Definition 1** ($\alpha$**-Bounded Collaborative Fairness**). *Client contributions (c) and rewards ($\theta^*$) are calculated based on two distinct performance metrics: for contributions, it is the performance of clients' standalone models (trained without collaboration); for rewards, it is the performance of the final models obtained after collaboration. Using the reward constraint $C_i < \theta_i^* \leq [(1-\alpha)\frac{(C_i)}{\max(C)} + \alpha]\max(\theta^*)$ as a foundation, fairness is quantified as $\gamma := 100 \times \rho(c, \theta^*)$, where $\rho()$ represents the Pearson Correlation Coefficient. A larger value of $\gamma$ indicates superior fairness of the framework.*

## 4 THE PROPOSED FEDRAC

In this section, we will introduce the details of the proposed FedRAC, a method that enhances CF by allocating appropriate rewards to clients while ensuring comprehensive model training. The architecture of FedRAC is shown in Figure 2, and its pseudo code is detailed in Appendix 1. First, we present the dynamic reputation calculation module in Section 4.1. Second, we introduce the rolling submodel allocation module in Section 4.2. Third, we propose the fairness guarantee theory in Section 4.3 to demonstrate that our proposed FedRAC is capable of achieving collaborative fairness. Finally, we perform a convergence analysis of FedRAC and establish its convergence properties in Section 4.4.

## 4.1 DYNAMIC REPUTATION CALCULATION

Client reputation plays a crucial role in determining model rewards and can evolve during training Xu et al. (2021). However, existing CF methods typically assign clients a fixed proportion of reputation-based rewards during training. These methods overlook the fact that the performance of the aggregated model gradually improves over time. As a result, low-contributing clients are under-rewarded in the early stages, hindering their full participation in training and leading to a decline in the aggregated model's performance.

To address the above issue, we propose a dynamic reputation calculation module designed to quantify the evolving reputation of clients, which dynamically aligns with their contributions during training. At the same time, this module enables clients with low contributions to access high-quality models in the early training stages, ensuring they can conduct adequate training without compromising the fairness. The module, illustrated in Figure 2, primarily consists of two steps. First, the

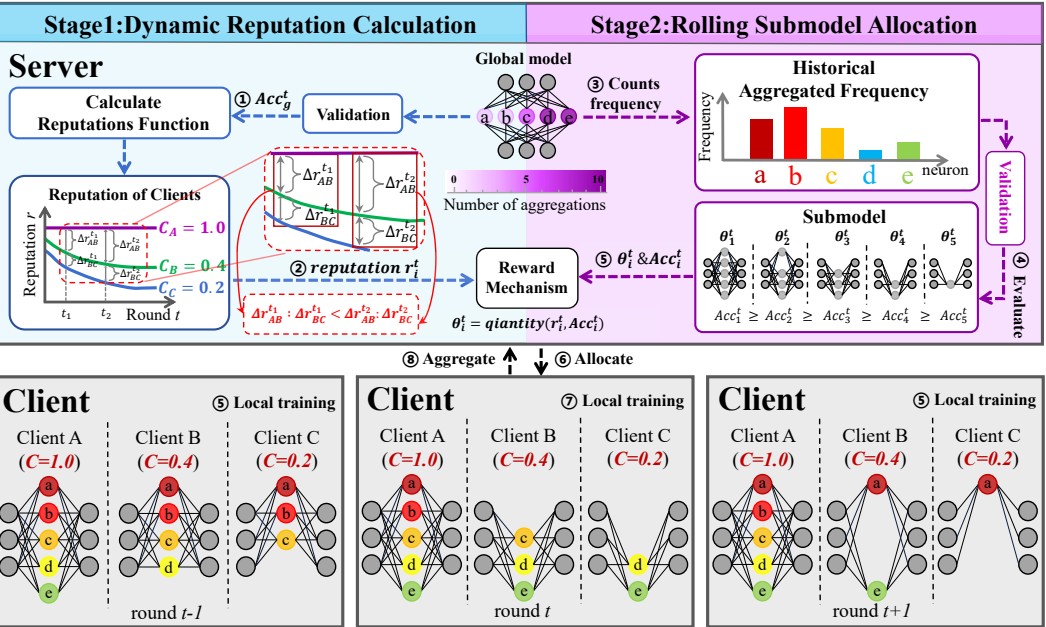

Figure 2: The overview of the FedRAC framework.

global model $\theta_g^t$ of the current round $t$ is validated on the server using the validation set. This process calculates the global model performance metric $Acc_g^t$. The step is meaningful because when the performance of $Acc_g^t$ is low, allocating excessive rewards to low-contribution clients will not impact the system's ultimate fairness.

Second, the client's reputation is derived via the corresponding dynamic reputation calculation function. Specifically, this function calculates a client's reputation based on its contribution and the current round's global model performance. The reputation evolves dynamically across training rounds. To quantify the client's information more fairly, we utilize the contribution $C_i$ of the client $i$ and normalize it to obtain the normalized contribution $C_i^n$:

$$C_i^n = \frac{e^{\beta * C_i}}{\max(e^{\beta * C})} * 100, \tag{2}$$

where $\beta$ is the hyperparameter. The equation (2) utilizes $\beta$ to provide a more differentiated representation of clients' contributions, thereby facilitating subsequent differentiated reward allocation. Then, we dynamically calculate the reputation $r_i^t$ of client $i$ in round $t$ based on their contributions $C_i$ and the performance of the aggregated model $Acc_g^t$:

$$r_i^t = C_i + (Acc_g^t - tmp_i^t) * C_i^n / 100, \tag{3}$$

$$tmp_i^t = Acc_g^t * \alpha * (1 - \frac{1}{log(\gamma * t + Con)}), \tag{4}$$

where $\alpha$ denotes the production parameter, and $Con$ is a constant. The equation 4 indicates that $tmp_i^t$ will increase as the training round $t$ and the aggregated model performance $Acc_g^t$ increase. From equation 3, we observe that in the early training stage, the reputation gap between clients' reputations is small, as the difference between the aggregated model performance $Acc_g^t$ and $tmp_i^t$ is slight. As training rounds progress, the reputation of low-contributing clients starts to decline. This gradually widens the gap with their high-contributing counterparts. The process continues until all clients' reputations converge to a fixed value. This approach ensures that low-contributing clients can receive more rewards in the early training stages without compromising fairness (i.e.,$\Delta r_{AB}^{t_1} : \Delta r_{BC}^{t_1} < \Delta r_{AB}^{t_2} : \Delta r_{BC}^{t_2}$). These results are shown in the Reputation of Clients Curve in Figure 2.

## 4.2 ROLLING SUBMODEL ALLOCATION

After obtaining each client's reputation (see Section 4.1), we assign the optimal submodel to clients based on their reputations to ensure system fairness. However, existing submodel assignment methods can lead to inter-model inconsistency (i.e., gradient-based) and intra-model inconsistency (i.e., submodel-based), both of which ultimately degrade the performance of the global model. For the former, using gradients as rewards tends to overlook the maintenance of consistency across local models; for the latter, assigning rewards based on the importance of submodel neurons can lead to inconsistent training frequencies among neurons, with some being trained too infrequently.

To address these issues, we proposed a rolling submodel allocation module. This module comprises two main steps: *Submodel Construction* and *Submodel Allocation*. The former ensures that each neuron in the global model is assigned the same number of times, while the latter guarantees each client receives a submodel matching their respective reputation. These two steps are elaborated in detail in the following sections.

### 4.2.1 SUBMODEL CONSTRUCTION

To ensure the consistency of neuron training frequency within the global model, this method is inspired by FedRolex Alam et al. (2022) and combines rolling submodel selection with fairness. Specifically, the server maintains a neuron frequency table $f^t$, such as the Historical Aggregation Frequency Table shown in Figure 2. This table is used to record the total historical aggregation times of all neurons up to the current round $t$:

$$f^0 = [0, 0, \ldots, 0] \in \mathbb{N}^K, \qquad (5a) \qquad \pi^t = \text{argsort}(f^{t-1}), \qquad (5b)$$

where $K$ denotes the neuron of the global model, $argsort(f)$ present soting $f$. In equation 5a, we initialize the table to $f^0$, which indicates that all neurons are assigned once at the initial stage. When allocating submodels in round $t$, we sort the neurons in ascending order according to the frequency table, as shown in formula (5b). The $\pi^t$ is a permutation, which arranges the neurons in round $t$ into index order from "least allocated" to "most allocated". Then, the server will prioritize neurons with fewer overall training times to form a series of new submodels based on the index order. We mask the neurons of the client $i$ as $m_{i,\pi}^t[j] \in \{0, 1\}^K$, where $j$ denotes the number of selected neurons. This indicates that in the current round, the neurons required by the client are selected in index order. The required neuron masks are $1$, and the unnecessary ones are $0$.

$$m_{i,\pi}^t[j] = \begin{cases} 1 & \pi \in \{\pi^t(1), \pi^t(2), \ldots, \pi^t(j)\}, \\ 0 & \text{otherwise.} \end{cases} \qquad (6a) \qquad f^t = f^{t-1} + \sum_{i \in S} m_i^t. \qquad (6b)$$

After all submodels for all clients are constructed, the neuron frequency table records the number of allocations in the current round based on formula (6b). Then, this process generates a new neuron frequency table $f^t$ for the next round. The series of submodels generated by the above method can ensure the consistency of the model architecture and avoid parameter imbalance training caused by random or static extraction.

### 4.2.2 SUBMODEL ALLOCATION

After generating the above series of submodels, their performance is evaluated to facilitate allocation based on clients' reputations. In specific implementation, we validate the global model $\theta_g^t$ and all submodels' masks $\sum_{i \in S} m_i^t$ on the validation set to obtain the performance $\sum_{i \in S} A_{K_i}^t$ of these submodels. Then, the corresponding submodel is assigned to the client $i$ based on the performances:

$$\theta_i^t = quantity(r_i, \sum_{i \in S} A_{K_i}^t), \qquad (7)$$

where $quantity(r_i, \sum_{i \in S} A_{K_i}^t)$ represents the submodel $\theta_i^t$ when $r_i = \sum_{i \in S} A_{K_i}^t$. $\theta_i^t$ denotes the submodel assigned to client $i$, and $r_i$ is calculated as detailed in Section 4.1. Allocating submodels in this way ensures that high-performance submodels are allocated to high-contributing clients.

$$\theta_g^{t+1} = \frac{\sum_{i \in S} \theta_i^t}{\sum_{i \in S} m_i^t(\theta_i^t, \theta_g^{t-1})}, \qquad (8)$$

Finally, after local training, all clients will upload their submodels $\theta_i^t$ along with their corresponding masks to the server. These are then used to generate the new aggregated model $\theta_g^{t+1}$ for the next round, as specified in equation (8). Subsequently, the server initiates the next training round based on the global model $\theta_g^{t+1}$ and the updated aggregation frequency $f^t$ of its internal neurons.

### 4.3 FAIRNESS ANALYSIS

In Section 4.1, we establish a mechanism to ensure clients with high contributions are assigned high reputations. Notably, the submodel $\theta_i$ allocated to client $i$ is determined based on the client's reputation $r_i$. Therefore, Clients with higher contributions thus receive better submodels. A key result of our work is that it guarantees a form of fairness under specific conditions associated with the loss function $F$. The proof is shown in Appendix A.

**Theorem 1** (**Fairness in Training Loss**). *FedRAC ensures collaborative fairness by allocating high-performance models to clients with high contributions as a form of reward. Formally, let $\delta_{i,t} := ||\theta_t - \theta_{i,t}||$. Assume that for $t \geq T$ ($T \in \mathbb{Z}^+$), $\theta_t$ is close to a stationary point of the loss function F, $F(\cdot)$ satisfies both L-smoothness and $\mu$-strong convexity with $L \leqslant \mu$. For any two clients $i, j \in \mathcal{N}$ in round t, if the reputation $r_i \geq r_j$, then $\theta_{i,t}$ is closer to $\theta_t$ than $\theta_{j,t}$ (i.e., $\delta_{i,t} \leqslant \delta_{j,t}$), which further implies $F(\theta_{i,t}) \leqslant F(\theta_{j,t})$.*

### 4.4 CONVERGENCE ANALYSIS

In this section, we will analyze the convergence of FedRAC. Since FedRAC ensures that every neuron in the global model is trained with equal frequency, the expected weight of client $i$'s submodel $\theta_i$ can be regarded as a contracted form of the global model $\theta_g$. Formally, $\theta_i^{t+1} = p_i\theta_g^t$, where $0 \leqslant p_i \leqslant 1$ denotes the contribution of client $i$. Consequently, equation (8) can be reformulated as the aggregation of submodels $\theta_i$, normalized by their corresponding $p_i$. Then, we introduce THEOREM 2, which establishes the convergence guarantee of FedRAC. The required Assumptions 3 and 4 follow prior studies Stich (2018); Stich et al. (2018); Yu et al. (2019). The proof is shown in Appendix B.

**Theorem 2** (**Asymptotic convergence**). *Under Assumptions 1 to 5, where $L$, $\mu$, $\sigma_i$, $G$, $p$ be defined. Set $\kappa = \frac{2}{\mu}$, $\gamma = max\{8\frac{L}{\mu}, E\} - 1$ and the learning rate $\eta_t = \frac{2}{\mu(\gamma+t)}$. Then FedRAC satisfies $E[F(\bar{\theta}_T)] - F^* \leqslant [\frac{L}{\gamma+T}(\frac{2B}{\mu^2} + \frac{\gamma+1}{2}\triangle_1)]$.*

## 5 EXPERIMENTS

In this section, we present extensive experiments to answer the following research questions: **RQl.** How does the fairness of our FedRAC compare to various state-of-the-art (SOTA) methods? **RQ2.** How does the predictive accuracy of our FedRAC differ from SOTA methods on multiple datasets? **RQ3.** How does the proportion of clients lying within the BCF interval (i.e., the rate) compare with that obtained by SOTA methods on different datasets? **RO4.** How do different components (i.e., dynamic reputation calculation module and rolling submodel allocation module) affect the fairness, the predictive model performance, and the rate?

### 5.1 SETUP

**Datasets and Models.** We evaluate FedRAC on three widely adopted public datasets in the collaborative fairness literature: CIFAR10 Krizhevsky et al. (2009), SVHN Netzer et al. (2011), and EMNIST Cohen et al. (2017). For all three datasets, we employ a feedforward neural network with two hidden layers. We construct four heterogeneous scenarios by adjusting dataset size and class distribution, namely POW (imbalanced data volume)(Xu et al. (2021)), CLA (imbalanced number of classes)(Lyu et al. (2020)), and DIR (imbalanced data volume and classes)(Chen et al. (2022); Gao et al. (2022); Yu et al. (2022); Yurochkin et al. (2019)), to evaluate CF under imbalanced data conditions. The detailed data split configurations are provided in Appendix D.1.

**Baselines.** We compare FedRAC with several representative federated learning methods, including FedAvg( McMahan et al. (2017)), CFFL( Lyu et al. (2020)), CGSV( Xu et al. (2021)), FedAVE( Wang et al. (2024d)), IAFL( Wu et al. (2024b)), and FedSAC( Wang et al. (2024e)), as well

| Dataset | CIFAR10 | | | | SVHN | | | | EMNIST | | | |
|---------|---------|-----|----------|----------|--------|--------|----------|----------|--------|--------|----------|----------|
| No. Clients | 10 | | | | 10 | | | | 10 | | | |
| Scene | POW | CLA | DIR(3.0) | DIR(7.0) | POW | CLA | DIR(3.0) | DIR(7.0) | POW | CLA | DIR(3.0) | DIR(7.0) |
| FedAvg | -17.99 | 74.86 | -1.51 | -10.80 | -33.93 | 85.15 | -1.04 | 56.81 | -66.62 | 18.96 | 7.46 | -58.19 |
| CFFL | 95.79 | 98.78 | 75.18 | 81.62 | 93.54 | 97.76 | 75.74 | 90.48 | 87.01 | 95.53 | 74.09 | 79.39 |
| CGSV | 75.75 | 98.64 | 75.30 | 90.56 | 96.91 | 95.20 | 91.21 | 92.91 | 93.39 | 96.91 | 75.93 | 81.60 |
| FedAVE | 86.39 | _99.33_ | 76.13 | 82.37 | 95.80 | 98.85 | 74.76 | 88.03 | 90.53 | 97.97 | 64.96 | 83.09 |
| IAFL | 98.33 | 99.28 | 88.01 | 97.56 | _99.45_ | _99.53_ | **98.23** | 96.61 | 97.05 | _99.11_ | 54.70 | 85.06 |
| FedSAC | _99.28_ | 96.67 | _98.50_ | _99.49_ | 99.05 | 96.32 | _96.33_ | _99.21_ | _98.90_ | 97.01 | _90.88_ | **93.72** |
| Ours | **99.34** | **99.72** | **98.71** | **99.56** | **99.84** | **99.86** | **98.23** | **99.66** | **99.23** | **99.32** | **93.62** | _93.57_ |

Table 1: **Comparison results of fairness** $\rho \in [-100, 100]$ with different methods on three datasets.

| Dataset | CIFAR10 | | | | SVHN | | | | EMNIST | | | |
|---------|---------|-----|----------|----------|--------|--------|----------|----------|--------|--------|----------|----------|
| No. Clients | 10 | | | | 10 | | | | 10 | | | |
| Scene | POW | CLA | DIR(3.0) | DIR(7.0) | POW | CLA | DIR(3.0) | DIR(7.0) | POW | CLA | DIR(3.0) | DIR(7.0) |
| Standalone | 40.63 | 36.65 | 31.48 | 38.68 | 68.35 | 59.11 | 61.43 | 63.18 | 71.05 | 64.23 | 78.60 | 78.63 |
| FedAvg | 48.21 | 42.16 | 51.57 | 52.48 | 78.58 | _75.13_ | _82.17_ | _82.21_ | 76.04 | 71.88 | **83.55** | _83.70_ |
| CFFL | 47.66 | 42.79 | 49.31 | 50.06 | 77.35 | 68.07 | 79.42 | 79.82 | 77.33 | 68.01 | _83.08_ | 80.94 |
| CGSV | 47.30 | 43.73 | 49.78 | 51.85 | 78.28 | 72.87 | 75.52 | 80.86 | 76.81 | 75.07 | 82.54 | 83.03 |
| FedAVE | 48.62 | 42.14 | 52.74 | _52.98_ | 77.61 | 60.89 | 71.24 | 78.48 | 78.84 | 73.79 | 78.14 | 80.01 |
| IAFL | 46.73 | 42.11 | 50.06 | 48.04 | 76.57 | 73.51 | 78.95 | 79.87 | 76.07 | 73.32 | 82.57 | 82.42 |
| FedSAC | _49.28_ | _44.24_ | _52.91_ | 52.42 | _78.98_ | 73.84 | 79.34 | 80.88 | _80.01_ | _76.53_ | 82.14 | 81.17 |
| Ours | **49.34** | **44.30** | **53.06** | **54.74** | **79.01** | **76.85** | **83.12** | **82.52** | **80.14** | **76.68** | **83.55** | **84.27** |

Table 2: **Comparison results of accuracy** with different methods on three datasets.

as a Standalone training setting where each client trains independently. We refer to Appendix D.2 for detailed descriptions and implementation.

**Hyper-Parameters.** All hyper-parameters are tuned via grid search under FedAvg(McMahan et al. (2017)). For fair comparison, we report the best-performing fairness metric achieved by each baseline method. Further details regarding the impact of hyper-parameter $\beta$ in FedRAC, along with additional configuration settings, are provided in the Appendix D.3.

**Implementation.** All experiments are run on a 64 GB-RAM Ubuntu 18.04.6 server with Intel(R) Xeon(R) CPU E5-2630 v4 @ 2.20GHz and 1 NVidia(R) 3090 GPUs.

## 5.2 EXPERIMENTAL RESULTS

**Fairness comparison (RQ1).** To evaluate the fairness of FedRAC, we conducted comparative experiments across three datasets (CIFAR10, SVHN, and EMNIST). Table 1 presents the fairness results of our method under different datasets and scenarios. FedRAC demonstrates significant advantages in all scenarios. In the POW, CLA, and DIR(3.0) settings on the SVHN dataset, it achieves fairness scores of 99.84%, 99.86%, and 98.23%, respectively, which are the highest in these scenarios. These results indicate that FedRAC effectively balances the contribution of each client, ensuring that clients with higher contributions obtain high-quality models. Moreover, in the POW and DIR(7.0) scenarios on the CIFAR10 dataset, FedRAC achieves fairness scores of 99.34% and 99.56%. These results significantly outperform other methods, with FedAvg recording the lowest score of -17.99%. In the CLA scenario of the EMNIST dataset, FedRAC achieves a fairness score of 99.32%, significantly outperforming methods such as CFFL (95.53%) and CGSV (96.91%). From these results, it is evident that FedRAC enhances client fairness through more reasonable reward distribution and strategic optimization, particularly DIR scenarios.

**Predictive performance (RQ2).** To evaluate the accuracy of FedRAC, we conducted a comparison with baseline methods. Table 2 presents the accuracy achieved by FedRAC across different datasets and scenarios. First, FedRAC outperforms FedAvg and other baseline methods in both the POW and DIR scenarios, demonstrating significant advantages. Specifically, in the CIFAR10 dataset, FedRAC achieves accuracies of 49.34% and 44.30% in the POW and CLA data splits, respectively, outperforming FedAvg (48.21% and 42.16%) and other baselines. On the SVHN dataset, FedRAC delivers the best performance, achieving the highest accuracy in all scenarios. Notably, in the DIR(7.0) scenario, FedRAC achieves an accuracy of 82.52%, significantly surpassing CGSV (80.86%) and FedAVE (78.48%). In the EMNIST dataset, FedRAC consistently maintains high accuracy across various data splits. For example, in the DIR(7.0) scenario, FedRAC achieves an accuracy of 84.27%, higher than FedAVE (80.01%), CFFL (80.94%), and FedSAC (81.17%). In short, compared to tra-

| Dataset | CIFAR10 | | | | SVHN | | | | EMNIST | | | |
|---|---|---|---|---|---|---|---|---|---|---|---|---|
| No. Clients | 10 | | | | 10 | | | | 10 | | | |
| Scene | POW | CLA | DIR(3.0) | DIR(7.0) | POW | CLA | DIR(3.0) | DIR(7.0) | POW | CLA | DIR(3.0) | DIR(7.0) |
| FedAvg | 0.1 | 0.1 | 0.1 | 0.1 | 0.1 | 0.1 | 0.1 | 0.1 | 0.4 | 0.4 | 0.1 | 0.2 |
| CFFL | 0.3 | 0.7 | 0.8 | 0.5 | 0.3 | 0.6 | 0.4 | 0.6 | 0.5 | 0.7 | 0.2 | 0.4 |
| CGSV | 0.4 | 0.9 | 0.8 | 0.8 | 0.4 | 0.4 | 0.9 | 0.6 | 0.5 | 0.2 | 0.2 | 0.3 |
| FedAVE | 0.3 | 0.8 | 0.5 | 0.8 | 0.4 | 0.7 | 0.3 | 0.3 | 0.4 | 0.3 | 0.3 | 0.5 |
| IAFL | 0.9 | 0.9 | **1.0** | **1.0** | 0.8 | 0.8 | 0.8 | 0.8 | 0.5 | **0.9** | 0.5 | 0.3 |
| FedSAC | **1.0** | **1.0** | **1.0** | **1.0** | **1.0** | 0.9 | **1.0** | **1.0** | 0.8 | **0.9** | 0.8 | 0.8 |
| Ours | **1.0** | 0.9 | **1.0** | **1.0** | **1.0** | **1.0** | 0.9 | **1.0** | **1.0** | **0.9** | **1.0** | **1.0** |

Table 3: **Comparison results of rate** with different methods on three datasets.

ditional methods, FedRAC not only ensures high-quality models for high-contribution clients but also consistently improves overall prediction performance across different data splits and scenarios. The overall performance comparison for achieving CF with SOTA methods on CIFAR10,SVHN,and EMNIST is provided in Appendix D.4.

**Rate (RQ3).** To further assess the fairness of FedRAC, we introduced the rate metric, which represents the proportion of clients whose reward allocation satisfies the BCF boundary. Table 3 shows the rates of methods satisfying the BCF boundary across different datasets and scenarios. It is evident that FedRAC consistently outperforms all other methods, achieving rates above 0.9 in most scenarios. On the CIFAR10, SVHN, and EMNIST datasets, it reaches a perfect rate of 1.0 in several cases, demonstrating full compliance with the BCF boundary. On the SVHN dataset, FedRAC achieves the highest performance in all scenarios. For example, in the CLA and DIR(7.0) splits, FedRAC achieves a perfect rate of 1.0, whereas FedAvg attains only 0.1. Meanwhile, in the EMNIST dataset, FedRAC consistently maintains the best results, achieving a rate of 1.0 in all scenarios. Table 3 validates the effectiveness of our approach. It shows that FedRAC ensures that reward allocation for clients in all scenarios complies with the BCF boundary, thereby guaranteeing CF.

**Ablation study(RQ4).** To evaluate the effectiveness of the two proposed modules in FedRAC, we conducted ablation studies on the SVHN dataset with 10 clients. $w/o\ reputation$ refers to statically calculating a client's reputation, where each client's contribution weight remains constant throughout training. Figure 3 illustrates the effect of dynamic reputation calculation, showing that it allows high-contribution clients to receive higher reputations, thereby improving overall model performance. For example, under the DIR(7.0) scenario on SVHN, accuracy drops from 82.52% to 70.56% without this module, and under the DIR(3.0) scenario, fairness decreases from 98.23% to 95.87%, highlighting its importance in maintaining both fairness and accuracy.

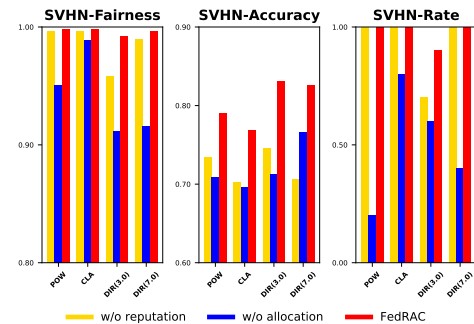

Figure 3: **Ablation studies** on FedRAC for fairness, accuracy, and rate on SVHN.

$w/o\ allocation$ refers to removing the dynamic submodel allocation, which randomly constructs client submodels. In the POW scenario on SVHN, , as shown in Figure 3,accuracy decreases from 79.01% to 70.81%, demonstrating that this module contributes to both fairness and stability. Overall, the ablation study confirms that both the two designed modules in FedRAC are critical for enhancing CF. A more detailed analysis of the CIFAR-10 dataset is provided in Appendix D.5.

# 6 CONCLUSION

In this work, we propose a novel FL framework, FedRAC, which achieves high-performance CF. It ensures that low-contribution clients receive appropriate rewards in the early stages and that all neurons in the aggregated model are trained evenly. The method also guarantees that high-contribution clients obtain a higher reputation, producing better sub-models and improving fairness, accuracy, and rate. Experiments show that FedRAC not only achieves superior fairness and accuracy across heterogeneous scenarios but also keeps all clients' rewards consistently within the BCF range. In future work, we plan to explore the potential of FedRAC for large-scale models.

ETHICS STATEMENT

This research does not involve human participants, animal experiments, or any other studies that require ethical approval. The datasets used in our experiments are publicly available and have been collected in accordance with privacy regulations. We have ensured that all personal data, if any, are anonymized before use. We have also considered the potential societal impact of our work and have taken measures to avoid harmful uses of the technology discussed.

REPRODUCIBILITY STATEMENT

The code used to conduct the experiments in this paper will be made publicly available upon publication. We provide detailed instructions on how to reproduce our results, including the training scripts, dataset access links, and environment specifications. We have made every effort to ensure that the code is well-documented and easy to use. The data used in our experiments are publicly available, and all experimental settings, including hyperparameters and evaluation metrics, are clearly detailed in the main text.

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

# A PROOF OF THEOREM 1

We let $\delta_{i,t} := ||\theta_t - \theta_{i,t}||$. If $r_i \geq r_j$, and the submodel $\theta_{i,t}$ obtained by client $i$ encompasses the submodel $\theta_{j,t}$ obtained by client $j$ ($\theta_{j,t} \in \theta_{i,t} \in \theta_t$). Then, the submodel $\theta_{i,t}$ obtained by client $i$ will exhibit closer alignment to the aggregated model $\theta_t$ in round $t$. It is obvious that $\delta_{i,t} \leqslant \delta_{j,t}$.

Subsequently, we use $\delta_{i,t} \leqslant \delta_{j,t}$ and some regularity conditions of $F()$ to establish $F(\theta_{i,t}) \leqslant F(\theta_{j,t})$. Specifically, we assume $F()$ is both $L$-smooth and $\mu$-strongly convex with $L \leqslant \mu$. We first recall the respective definitions for $L$-smooth functions and $\mu$-strongly convex.

**Assumption 1** ($L$-**smooth F**). *If $F$ is $L$-smooth, then $\forall \theta_i, \theta_j \in \theta$,*

$$F(\theta_i) \leqslant F(\theta_j) + \nabla F(\theta_j)^T(\theta_i - \theta_j) + \frac{L}{2}||\theta_i - \theta_j||^2. \tag{9}$$

**Assumption 2** ($\mu$-**strongly convex F**). *If $F$ is $\mu$-strongly convex, then $\forall \theta_i, \theta_j \in \theta$,*

$$F(\theta_i) \geq F(\theta_j) + \nabla F(\theta_j)^T(\theta_i - \theta_j) + \frac{\mu}{2}||\theta_i - \theta_j||^2. \tag{10}$$

From $L$-smoothness, we have

$$F(\theta_{i,t}) \leqslant \underbrace{F(\theta_{N,t}) + \nabla F(\theta_{N,t})^T(\theta_{i,t} - \theta_{N,t}) + \frac{L}{2}\delta_{i,t}^2}_{R_L}. \tag{11}$$

From $\mu$-strongly convex, we have

$$F(\theta_{j,t}) \geq \underbrace{F(\theta_{N,t}) + \nabla F(\theta_{N,t})^T(\theta_{j,t} - \theta_{N,t}) + \frac{\mu}{2}\delta_{j,t}^2}_{R_\mu}. \tag{12}$$

In order to prove $F(\theta_{i,t}) \leqslant F(\theta_{j,t})$, it suffices to prove $R_L \leqslant R_\mu$ or equivalently $R_L - R_\mu \leqslant 0$.

$$R_L - R_\mu = \underbrace{\nabla F(\theta_{N,t})^T(\theta_{i,t} - \theta_{j,t})}_{R_1} + \underbrace{\frac{1}{2}(L\delta_{i,t}^2 - \mu\delta_{j,t}^2)}_{R_2}. \tag{13}$$

With $L \leqslant \mu$ and $\delta_{i,t} \leqslant \delta_{j,t}$, we have

$$R_2 = \frac{1}{2}(L\delta_{i,t}^2 - \mu\delta_{j,t}^2) \leqslant \frac{L}{2}(\delta_{i,t}^2 - \delta_{j,t}^2) \leqslant 0. \tag{14}$$

We define $\theta_{N,t}$ being close to a stationary point of F by establishing an upper limit on the gradient:

$$||\nabla F(\theta_{N,t})|| \leqslant \frac{L|\delta_{i,t}^2 - \delta_{j,t}^2|}{2||\theta_{i,t} - \theta_{j,t}||}. \tag{15}$$

We have the following:

$$|R_1| \triangleq |\nabla F(\theta_{N,t})^T(\theta_{i,t} - \theta_{j,t})|$$

$$\leqslant ||\nabla F(\theta_{N,t})|| \times ||(\theta_{i,t} - \theta_{j,t})||$$

$$\leqslant \frac{L|\delta_{i,t}^2 - \delta_{j,t}^2|}{2}$$

$$\leqslant |R_2|$$

where the first inequality is derived from the Cauchy-Schwarz, the second inequality is by substituting the aforementioned upper limit (refer to e equation (15)), and the last inequality emerges from taking the absolute values of two negative values (refer to equation (14)).

Finally, given that $|R_1| \leqslant |R_2|$ and $R_2 \leqslant 0$, we derive $R_1 + R_2 \leqslant 0$. Therefore, it follows that $R_L - R_\mu \triangleq R_1 + R_2 \leqslant 0$, which subsequently implies $F(\theta_{i,t}) \leqslant F(\theta_{j,t})$.

# B  PROOF OF THEOREM 2

Let $I_E$ be the set of global synchronization steps, i.e., $I_E = \{nE|n = 1, 2, ...\}$. For convenience, we define $v_i^{t+1}$ as the immediate result of one step SGD update from $\theta_i^t$, i.e., $v_i^{t+1} = \theta_i^t - \eta_t \nabla F_i(\theta_i^t, \xi_i^t)$. $\bar{g}_t = \sum_{i=1}^N \frac{\nabla F_i(\theta_i^t)}{p_i}$ and $g_t = \sum_{i=1}^N \frac{\nabla F_i(\theta_i^t, \xi_i^t)}{p_i}$. Therefore, $\bar{v}_{t+1} = \bar{\theta}_t - \eta_t g_t$ and $Eg_t = \bar{g}_t$.

**Assumption 3.** *Let $\xi_i^t$ denote samples uniformly from the local data of the $i$-th device at random. It is asserted that the variance of stochastic gradients within each device remains constrained:*

$$E\|\nabla F_i(\theta_i^t, \xi_i^t) - \nabla F_i(\theta_i^t)\| \leqslant \sigma_i^2, \tag{16}$$

**Assumption 4.** *The expected squared norm of stochastic gradients is uniformly constrained:*

$$E\|\nabla F_i(\theta_i^t, \xi_i^t)\| \leqslant G^2, \tag{17}$$

*where $i \in \{1, 2, ..., N\}$ and $t \in \{1, 2, ..., T - 1\}$.*

**Assumption 5.** *Each neuron in the aggregation model is assigned the same number of times after $T$ rounds. Therefore, the expected weight of the allocated submodel $\theta_i$ is a contraction of the aggregate model $\theta_g$, i.e., $\theta_i^{t+1} = p_i \theta_g^t$. Here, $p_i$ ($0 \leqslant p_i \leqslant 1$) denotes the long-term expectation of the size ratio between the submodel $i$ and the aggregate model obtained in multiple iterations, i.e., $\theta_g^{t+1} = \sum_{i=1}^N \frac{\theta_i^{t+1}}{p_i}$.*

**Lemma 1.** *(Result of one step SGD). Assume ASSUMPTION 1 and ASSUMPTION 2. If $\eta_t \leqslant \frac{1}{4L}$, we have*

$$E\|\bar{v}_{t+1} - \theta^*\|^2 \leqslant (1 - \eta_t \mu) E\|\bar{\theta}_t - \theta^*\|^2 + \eta_t^2 E\|g_t - \bar{g}_t\|^2 \\ + 6L\eta_t^2 \Gamma + 2E \sum_{i=1}^N \frac{\|\bar{\theta}_t - \theta_i^t\|^2}{p_i}, \tag{18}$$

where $\Gamma = F^* - \sum_{i=1}^N \frac{F_i^*}{p_i}$. LAMMA 1 has been made by Li et al. (2019).

For Assumption 3, the variance of stochastic gradients within device $i$ is constrained by $\sigma_i^2$. Consequently,

$$E\|g_t - \bar{g}_t\|^2 = E\|\sum_{i=1}^N \frac{1}{p_i}(\nabla F_k(\theta_i^t, \xi_i^t) - \nabla F_i(\theta_i^t))\|^2 \\ = \sum_{i=1}^N \frac{1}{p_i^2} E\|\nabla F_i(\theta_i^t, \xi_i^t) - \nabla F_i(\theta_i^t)\|^2 \\ \leqslant \sum_{i=1}^N \frac{1}{p_i^2} \sigma_i^2. \tag{19}$$

As FedRAC requires communication each $E$ steps. We let $\eta_t \leqslant 2\eta_{t+E}$. Therefore, for any $t \geq 0$, there exists a $t_0 \leqslant t$, such that $t - t_0 \leqslant E - 1$ and $\theta_i^{t_0} = \bar{\theta}_{t_0}$ for all $k = 1, 2, ..., N$. Then

$$E \sum_{i=1}^{N} \frac{1}{p_i} \|\bar{\theta}_t - \theta_i^t\|^2 = E \sum_{i=1}^{N} \frac{1}{p_i} \|(\theta_i^t - \bar{\theta}_{t_0}) - (\bar{\theta}_t - \bar{\theta}_{t_0})\|^2$$

$$\leqslant E \sum_{i=1}^{N} \frac{1}{p_i} \|\theta_i^t - \bar{\theta}_{t_0}\|^2$$

$$\leqslant E \sum_{t=t_0}^{t-1} (E-1)\eta_t^2 \|\nabla F_k(\theta_i^t, \xi_i^t)\|^2 \qquad (20)$$

$$\leqslant \sum_{t=t_0}^{t-1} (E-1)\eta_{t_0}^2 G^2$$

$$\leqslant \eta_{t_0}^2 (E-1)^2 G^2$$

$$\leqslant 4\eta_t^2 (E-1)^2 G^2.$$

We use $E\|X - EX\|^2 \leqslant E\|X\|^2$ where $X = \theta_i^t - \bar{\theta}_{t_0}$ with probability $\frac{1}{p_i}$. We use Jensen inequality:

$$\|\theta_i^t - \bar{\theta}_{t_0}\| = \|\sum_{t=t_0}^{t-1} \eta_t \nabla F_i(\theta_i^t, \xi_i^t)\|^2$$

$$\leqslant (t-t_0) \sum_{t-t_0}^{t-1} \sum_{t-t_0}^{t-1} \eta_t^2 \|\nabla F_i(\theta_i^t, \xi_i^t)\|^2. \qquad (21)$$

Here, we utilize $\eta_t \leqslant \eta_{t_0}$ for $t \geq t_0$ and $E\|\nabla F_k(\theta_i^t, \xi_i^t)\|^2 \leqslant G^2$ for $i = 1, 2, ..., N$. We $\eta_{t_0} \leqslant 2\eta_{t_0+E} \leqslant 2\eta_t$ for $t_0 \leqslant t \leqslant t_0 + E$.

Let $\triangle_t = E\|\bar{\theta}_t - \theta^*\|$. From equation 18, equation 19, and equation 20, it follows that

$$\triangle_{t+1} \leqslant (1 - \eta_t \mu)\triangle_t + \eta_t^2 \sum_{i=1}^{N} \frac{\sigma^2}{p_i^2} + 6L\eta_t^2 \Gamma + 8\eta_t^2 (E-1)^2 G^2$$

$$\leqslant (1 - \eta_t \mu)\triangle_t + \eta_t^2 \underbrace{(\sum_{i=1}^{N} \frac{\sigma^2}{p_i^2} + 6L\Gamma + 8(E-1)^2 G^2)}_{B} \qquad (22)$$

For a diminishing stepsize, $\eta_t = \frac{\kappa}{t+\gamma}$ for some $\kappa > \frac{1}{\mu}$ and $\gamma > 0$ such that $\eta_1 \leqslant min\{\frac{1}{\mu}, \frac{1}{4L}\} = \frac{1}{4L}$ and $\eta_t \leqslant 2\eta_{t+E}$. We will prove $\triangle \leqslant \frac{v}{\gamma+t}$ by induction, where $v = max\{\frac{\kappa^2 B}{\kappa\mu-1}, (\gamma+1)\triangle_1\}$. Firstly, the definition of $v$ guarantees its applicability for $t = 1$. Assuming the conclusion holds for some $t$, it follows that

$$\triangle_{t+1} \leqslant (1 - \eta_t \mu)\triangle_t + \eta_t^2 B$$

$$\leqslant (1 - \frac{\kappa\mu}{t+\gamma})\frac{v}{t+\gamma} + \frac{\kappa^2 B}{(t+\gamma)^2}$$

$$= \frac{t+\gamma-1}{(t+\gamma)^2} v + [\frac{\kappa^2 B}{(t+\gamma)^2} - \frac{\kappa\mu-1}{(t+\gamma)^2} v]$$

$$\leqslant \frac{t+\gamma-1}{(t+\gamma)^2} v + \frac{\kappa^2 B}{(t+\gamma)^2} - \frac{\kappa^2 B}{(t+\gamma)^2} \underbrace{- \frac{\kappa\mu-1}{(t+\gamma)^2 (\gamma+1)\triangle_1}}_{\leqslant 0} \qquad (23)$$

$$\leqslant \frac{v}{t+\gamma-1}$$

Then by the $L$-smoothness (ASSUMPTION 1) of $F$,

$$E[F(\bar{\theta}_T)] - F^* \leqslant (\bar{\theta}_T - \theta^*)^T \underbrace{\nabla F_i(\theta^*)}_{=0} + \frac{L}{2}\|\bar{\theta}_T - \theta^*\|_2^2$$

$$= \frac{L}{2}\triangle_T \tag{24}$$

$$\leqslant \frac{L}{2}\frac{v}{\gamma + T}$$

We let $\kappa = \frac{2}{\mu}, \gamma = max\{8\frac{L}{\mu}, E\} - 1$. In the lines 1293, we have

$$v = max\{\frac{\kappa^2 B}{\kappa\mu - 1}, (\gamma + 1)\triangle_1\}$$

$$\leqslant \frac{\kappa^2 B}{\kappa\mu - 1} + (\gamma + 1)\triangle_1 \tag{25}$$

$$\leqslant \frac{4B}{\mu^2} + (\gamma + 1)\triangle_1$$

Substituting equation 25 into equation 24, we obtain

$$0 \leqslant \lim_{T\to\infty} E[F(\bar{\theta}_T)] - F^* \leqslant \lim_{T\to\infty}[\frac{L}{\gamma + T}(\frac{2B}{\mu^2} + \frac{\gamma + 1}{2}\triangle_1)] = 0 \tag{26}$$

Therefore, $\lim_{T\to\infty} E[F(\bar{\theta}_T)] - F^* = 0$.

## C  ALGORITHM

---
**Algorithm 1** FedRAC
---
1: **Input:** The global model $\theta_g$, the local submodel $\theta_i$, submodels' performance $A_{K_i}$, the number of local update steps $E$, learning rate $\eta_t$, number of clients $N$, amplification factor $\alpha$, hyperparameter $\beta$, client's contribution $c$, neuron frequency table $f$
2: Initialize the global model parameters $\theta_g^0$ and neuron frequency table $f^0$
3: **for** round $t = 0, 1, \ldots, T - 1$ **do**
4:     Compute the normalized contribution $C_i^n$ (2) and the intermediate variable $tmp_i^t$ (4)
5:     Calculate the reputation $r_i^t$ of client $i$: $r_i^t = C_i + (Acc_g^t - tmp_i^t) * C_i^n/100$
6:     Submodel $i$'s mask in round $t$: $m_i^t$ (6a) and upgrade frequency table $f^t$ (6b)
7:     Calculate Submodels $\theta_i^t$ of clients $i$ in round $t$: $\theta_i^t = quantity(\gamma_i, \sum_{i \in S} A_{K_i}^t)$
8:     **for** each client $i \in N$ **do**
9:         **for** each iteration $j = 0, 1, \ldots, E - 1$ **do**
10:             $\theta_{i,j+1} \leftarrow \theta_{i,j} - \eta_t\nabla F_i(\theta_{i,j})$
11:         **end for**
12:     **end for**
13:     The server aggregates the received submodels: $\theta_g^{t+1} = \frac{\sum_{i \in S} \theta_i^t}{\sum_{i \in S} m_i^t(\theta_i^t, \theta_g^{t-1})}$
14: **end for**
---

This is the pseudo code of FedRAC.The algorithm first initializes the global model parameters and the neuron frequency table, followed by executing the subsequent steps in each training round: computing standardized reputation based on clients' contributions to ensure that high-contribution clients receive higher reputation while enabling low-contribution clients to obtain appropriate rewards in the early stages of training; subsequently, generating submodel masks and updating the neuron frequency table to guarantee balanced allocation of all neurons in the global model; then, each client performs multiple local training iterations on their local data to update submodel parameters; finally, the server updates the global model parameters through weighted aggregation of the clients' submodels.

# D    ADDITIONAL EXPERIMENTS

## D.1    DATA SPLITS

**POW (Power-law data volume).** We apply a power-law distribution to randomly assign the entire dataset to multiple clients, ensuring imbalanced data volume. For example, in CIFAR10 with 20,000 samples distributed across 10 clients, clients with larger data volumes are expected to achieve better predictive performance.

**CLA (Class imbalance).** We vary the number of classes while keeping the total data volume unchanged. In a five-client setup using CIFAR10, Clients 1, 2, 3, 4, and 5 are assigned local training data containing 1, 3, 5, 7, and 10 classes, respectively.

**DIR (Dirichlet imbalance).** We adopt a Dirichlet distribution $DIR(\alpha)$ to provide each client with data of varying volumes and classes. Specifically, we sample $p_i^l \sim DIR(\alpha)$ from a Dirichlet distribution with parameter $\alpha$, and assign class $l$ to client $i$ according to the sampled proportions $p_i^l$.

## D.2    BASELINE DESCRIPTIONS

We provide detailed descriptions of all baseline methods considered in our experiments. FedAvg distributes the same model to all clients at each communication round. CFFL computes a reputation score based on local accuracy and data size (or label diversity), allocating more gradients to clients with higher reputations. CGSV rewards clients whose local model gradients are more similar to the global gradient. FedAVE compares the similarity between local and ideal data distributions, assigning more gradients to clients with higher similarity. IAFL allocates more gradient updates to highly contributive clients while distributing a reference model randomly to all clients. FedSAC evaluates neuron importance in the global model and assigns more important neurons to clients with higher contributions. Standalone denotes the case where each client trains its model independently without federated aggregation. To ensure fairness, we require all algorithms to allocate rewards according to client contributions, rather than relying on their original reputation-based mechanisms.

## D.3    HYPERPARAMETER SETTINGS

In Table 4, we present the performance of Fe-dRAC under different values of $\beta$ across the POW, DIR(3.0), and DIR(7.0) scenarios of CIFAR10. The experimental results show that as $\beta$ increases, the maximum test accuracy gradually decreases. This is because as $\beta$ decreases, the size of the submodels downloaded by clients increases. When $\beta$ is small, the submodels of low-contribution clients contain

| Scene | POW | DIR(3.0) | DIR(7.0) |
|---|---|---|---|
| FedRAC ($\beta = 0.5$) | **50.11** | **53.15** | **54.37** |
| FedRAC ($\beta = 1$) | 49.34 | 53.01 | 54.06 |
| FedRAC ($\beta = 2$) | 48.99 | 52.89 | 54.02 |
| FedRAC ($\beta = 3$) | 48.91 | 52.45 | 53.67 |

Table 4: **Comparison results of FedRAC with different $\beta$ values** on various scenes.

more neurons, enabling effective training to enhance the performance of all local models.

## D.4    OVERALL PERFORMANCE IN COLLABORATIVE FAIRNESS

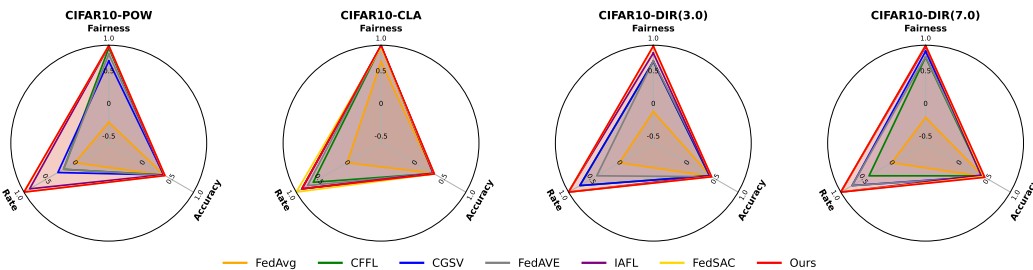

Figure 4: CIFAR10's visualization results of fairness, accuracy, and rate.

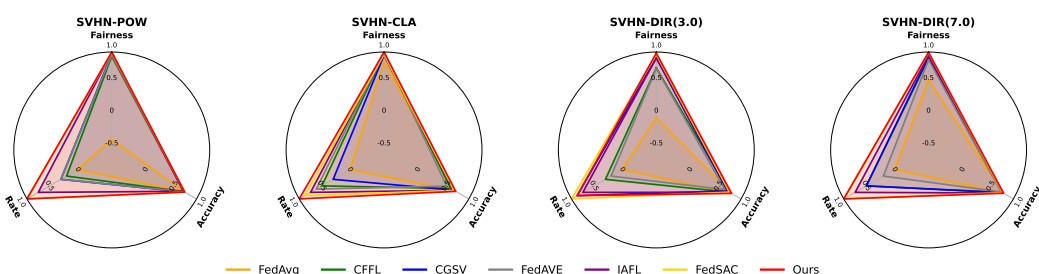

Figure 5: SVHN's visualization results of fairness, accuracy, and rate

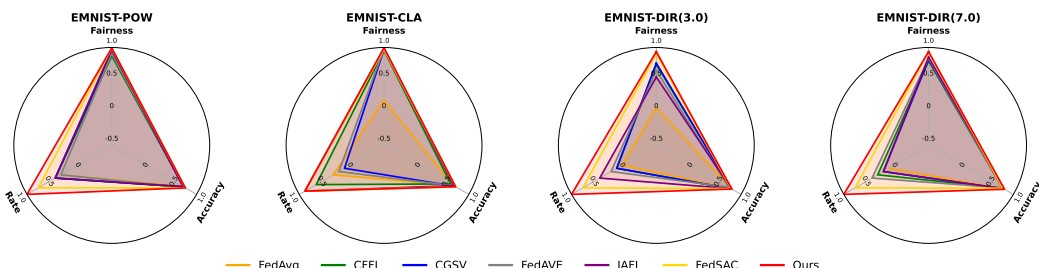

Figure 6: EMNIST's visualization results of fairness, accuracy, and rate

The above three figures illustrate the visualization results of fairness, accuracy, and rate across different datasets: CIFAR10, SVHN, and EMNIST. Each dataset presents the performance of various methods (FedAvg, CFFL, CGSV, FedAVE, IAFL, FedSAC, and Ours) under four different data splits (POW, CLA, DIR(3.0), and DIR(7.0)). In CIFAR10, fairness, accuracy, and rate distributions across scenarios show that FedRAC consistently maintains high performance. SVHN yields similar results, particularly in extreme cases such as DIR(7.0), while EMNIST further corroborates FedRAC's superiority. These visualizations provide an intuitive comparison of different methods across multiple dimensions.

## D.5 ABLATION STUDY

In the SVHN dataset, we analyzed the performance under four different splits: POW, CLA, DIR(3.0), and DIR(7.0). For each split, three key metrics were recorded: accuracy, fairness, and rate. Specifically, the accuracy in SVHN varied across scenarios. For example, under the DIR(7.0) scenario, accuracy dropped from 82.52% to 70.56%. Fairness showed only slight fluctuations, remaining relatively stable with values of 95.87% in DIR(3.0) and 99.01% in DIR(7.0). The rate varied more noticeably across splits, ranging from 0.7 to 1.0, reflecting differences in performance across scenarios.

In the CIFAR10 dataset, similar trends were observed. For the splits POW, CLA, DIR(3.0), and DIR(7.0) in CIFAR10, accuracy, fairness, and rate were calculated. In the DIR(3.0) scenario, the accuracy was 50.22%, and fairness decreased from 98.71% to 88.17%, indicating a larger impact on fairness when the Dynamic Reputation Calculation module was removed. For the rate, CIFAR10 performed optimally under the DIR(7.0) scenario with a value of 1.0. Additionally, FedRAC consistently achieved the best results across all splits and scenarios, particularly in the CIFAR10 POW scenario, where accuracy reached 49.34%, fairness achieved 99.34%, and the rate was 1.0, clearly outperforming both $w/o\ reputation$ and $w/o\ allocation$.

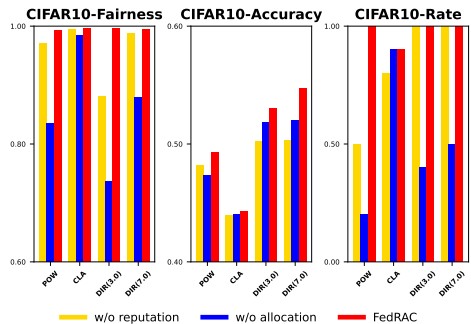

Figure 7: The ablation study in CIFAR10.

These ablation studies confirm that both core modules in FedRAC are indispensable. The Dynamic Reputation Calculation module is essential for maintaining fairness and accuracy, while the Rolling

Submodel Allocation module further enhances performance and stability, allowing FedRAC to significantly outperform all baselines in both fairness and accuracy.

## E USE OF LLMS

In this research, we used large language models (LLMs) such as GPT-5 to assist with text refinement and improve the clarity and readability of the manuscript. The LLMs were primarily employed for grammar correction, style suggestions, and rephrasing of sentences, ensuring the manuscript's language quality. It is important to note that the use of LLMs was limited to enhancing the language and presentation, and all research ideas, results, and analysis were independently conducted by the authors. No LLMs were used to generate content or make any intellectual contributions to the core research.

