# OpenReview forum: "FedRAC: Rolling Submodel Allocation for Collaborative Fairness in Federated Learning"
_ICLR.cc/2026/Conference — ICLR 2026 Conference Withdrawn Submission_

### Official Review · Reviewer_ijSQ · 2025-10-28

**Soundness:** 2
**Presentation:** 2
**Contribution:** 2
**Rating:** 4
**Confidence:** 3

**Summary:**

This paper proposes a collaborative fairness-aware federated learning framework named FedRAC. The framework introduces a dynamic reputation calculation module that adaptively updates clients’ reputations during training, allowing low-contributing clients to access better models in the early stages. In addition, it designs a rolling submodel allocation module that allocates high-performance submodels to clients with higher reputations, ensuring all neurons in the global model are effectively trained.

**Strengths:**

1. **Meaningful research topic:** Fairness is an important and practical issue in federated learning, making the research problem well-motivated.
2. **Theoretical support:** The proposed framework is accompanied by theoretical analysis.

**Weaknesses:**

1. **Unclear explanation of motivations:** The paper mentions that existing static reputation approaches lead to low-contributing clients being under-rewarded in the early stages. However, it is unclear what specific static reputation methods are being referred to, why they would cause such an issue, and how this under-rewarding affects the final model performance. The purpose and meaning of Figure 1(a) are not clearly explained, making it difficult to understand how it supports the argument.
2. **Limited discussion of the “intra-modal inconsistency” problem:** It seems that this problem mainly arises in submodel-based methods, but the paper does not clarify whether such methods are mainstream in fairness-aware FL.
3. **Organization and emphasis:** The advantage of assigning higher reputations to low-contributing clients in the early stages is not well highlighted in the introduction or motivation sections, which weakens the logical flow of the paper.
4. **Experimental setup limitations:** The datasets used are relatively simple, the number of clients is small, and the performance improvement over baselines is modest. These factors limit the empirical validation of the proposed approach.
5. **Unclear validation setting:** The paper mentions that submodel performance is evaluated on a validation set, but it is unclear how this validation set is defined, whether it represents the overall data distribution, and whether such validation results align with clients’ expectations.

**Questions:**

1.	Which specific “static reputation methods” are being referred to in Figure 1(a)? Why they would cause low-contributing clients to be under-rewarded in the early stages? Additionally, how does this under-rewarding affect the overall model performance?
2.	Is the “intra-modal inconsistency” problem unique to submodel-based methods? Are these methods considered the mainstream approach in fairness-aware FL?
3.	How is the validation set constructed for submodel performance evaluation? How is its representativeness ensured? How to guarantee the validation results align with clients’ expectations?
4.	The experimental results show only limited improvement. Can the authors provide more challenging datasets or larger-scale experiments to strengthen the empirical evidence?

---

### Official Review · Reviewer_wHrN · 2025-10-30

**Soundness:** 2
**Presentation:** 2
**Contribution:** 2
**Rating:** 4
**Confidence:** 3

**Summary:**

This paper proposes FedRAC, a federated learning (FL) framework designed to address collaborative fairness—ensuring clients are rewarded for their contributions to sustain long-term participation. The method  designs a dynamic reputation calculation module, un-
der a theoretical fairness guarantee.  Comprehensive experiments on three datasets (CIFAR-10, SVHN, and EMNIST) to demonstrate FedRAC’s superiority over state-of-the-art methods in both accuracy and fairness.

**Strengths:**

1. propose a novel FL framework, FedRAC, which achieves high-performance in both fairness and accuracy.
2. theoretical guarantees are provided in the paper, though I did not validate the meth details

**Weaknesses:**

1. Some part of the paper is hard to follow. For example, what is the triangle with subscript 1 in Theorem 2? What is the fairness metric in Section 5.2? Figure 3 is a bit hard to read.
2. The main results in Table 1 to 3 could have standard errors, so that readers know whether the differences are significant

**Questions:**

see weaknesses

---

### Official Review · Reviewer_ntxN · 2025-11-01

**Soundness:** 3
**Presentation:** 3
**Contribution:** 2
**Rating:** 4
**Confidence:** 3

**Summary:**

FedRAC introduces a collaborative-fairness FL framework that couples a dynamic, theoretically fair reputation mechanism with a rolling submodel allocation strategy. Reputations diverge over time so low-contribution clients can access stronger models early, while high-reputation clients later receive higher-performing submodels. The allocator prioritizes low-frequency neurons to even out training across the global model and claims convergence guarantees. Experiments on CIFAR-10, SVHN, and EMNIST report gains in both fairness and accuracy over prior gradient- and submodel-based baselines.

**Strengths:**

Thoughtful integration of fairness and accuracy: dynamic reputations mitigate early under-rewarding, and rolling submodel allocation explicitly targets inter- and intra-model inconsistency with theoretical backing.

**Weaknesses:**

1. The functional form of tmp_i^t in Equation (4) appears arbitrary and lacks intuitive or theoretical motivation. The authors should provide a stronger justification for this specific form and compare it with alternative designs.

2. The paper does not discuss the computational and communication costs involved in maintaining the neuron frequency table, sorting it, and evaluating all submodels each round. This oversight is critical for assessing FedRAC’s scalability, particularly for larger models.

3. The function \text{quantity}(r_i, \sum A_{K_i}^t) in Equation (7) is not clearly defined. The mapping from continuous reputation to discrete submodel selection requires a more precise algorithmic description to ensure reproducibility.

4. The manuscript contains several grammatical errors and awkward phrasings (e.g., ``present soting’’). Professional proofreading is recommended to improve clarity and readability.

5. Tracking neuron-level frequencies and performing rolling submodel extraction/allocation may incur significant scheduling, compute, and communication costs—especially for large or sparse models.

6. The fairness/convergence guarantees may rely on idealized conditions (e.g., synchronized rounds, bounded heterogeneity, non-adversarial updates) and could weaken under severe non-IID data, partial participation, or label noise.

7. Dynamic reputations and reward allocation could be gamed (e.g., strategic updates/poisoning to gain better submodels), and early up-weighting of low-contributors might temporarily penalize high-quality contributors.

**Questions:**

1. What are the exact assumptions behind the fairness and convergence proofs, and how do results change under extreme non-IID distributions, intermittent participation, or noisy labels?

2. What is the measured computation/communication overhead of rolling submodel allocation and neuron-frequency tracking, and how does it scale to modern large models (e.g., ViTs, LLMs)?

3. How does FedRAC mitigate strategic behavior or adversarial clients (poisoning/backdoor attempts) that could exploit dynamic reputations to obtain stronger submodels without legitimate contribution?

4. Could you provide more intuition or theoretical justification behind the design of tmp_i^t  in Equation (4)? Have you explored alternative forms (e.g., linear decay, exponential decay)? A figure illustrating its behavior over rounds would be helpful for clarity.

5. How does FedRAC scale with increasing model size or number of clients? Could you include an analysis or experimental results comparing training time, memory usage, or communication costs against baselines like FedSAC?

6. Can you clarify the exact mechanism of the quantity(·) function in Equation (7)? Is it a lookup, interpolation, or binning strategy? A more detailed algorithmic description would improve reproducibility.

7. The statement that "when Acc_g^t is low, allocating excessive rewards to low-contribution clients will not impact the system's ultimate fairness" seems overly strong. Could you qualify this claim with more nuance or empirical support?

---

### Official Review · Reviewer_h4Js · 2025-11-12

**Soundness:** 3
**Presentation:** 3
**Contribution:** 2
**Rating:** 2
**Confidence:** 3

**Summary:**

In this paper, the authors focus on the collaborative fairness problem in federated learning, in which clients should be rewarded proportionally to their contribution. It was discussed about two identified sources of performance degradation. To address these, FedRAC is proposaed with two modules, the Dynamic Reputation Calculation module and the Rolling Submodel Allocation module. It has presented experiments on CIFAR‑10, SVHN, and EMNIST across multiple heterogeneous splits (POW, CLA, DIR(3.0/7.0)). The results show improved fairness (Pearson correlation between contribution and reward) and test accuracy over baselines (FedAvg, CFFL, CGSV, FedAVE, IAFL, FedSAC). An additional rate metric (proportion of clients whose reward lies in an α‑bounded CF interval) is reported.

**Strengths:**

**1. Originality**

> My understanding of this work is that, it has combined reputation dynamics with rolling submodel allocation, drawing inspiration from model-heterogeneous FL (e.g., FedRolex) while aiming at collaborative fairness, not just resource heterogeneity. This connection between fairness-aware reward policies and rolling extraction is a useful design angle that, to my knowledge, is not directly explored in earlier CF methods.

> The idea to prioritize low-frequency neurons to mitigate intra-model inconsistency is a plausible enhancement over static/random extraction and directly addresses known weaknesses of submodel-based approaches.

**2. Quality**

> The problem is well-motivated within the CF literature (CFFL, CGSV, FedAVE) and distinguishes itself from group fairness (FairFed) and robustness/shapley-based contribution estimation. The baseline coverage thus spans gradient- and submodel-based CF methods.

> The empirical section is systematic: three datasets (CIFAR‑10, SVHN, EMNIST), four heterogeneity regimes (POW/CLA/DIR), three metrics (fairness, accuracy, rate), and ablations on both core modules. The claimed improvements are consistent across settings.

**3. Clarity**

> The paper explains where existing methods fail (static reputations, inter-/intra-model inconsistencies) with clear illustrations (Fig. 1) and provides an architecture diagram (Fig. 2) that makes the two-stage pipeline easy to follow.

> Definitions of α‑bounded collaborative fairness and the rate metric are explicit, and experimental setups (data splits, hardware) are documented.

**4. Significance**

> Collaborative fairness is timely and distinct from client/group fairness and robustness; recent surveys emphasize the importance of fairness dimensions in FL beyond accuracy. The paper improves CF while not sacrificing accuracy, which is often hard in practice.

> The design space explored here (dynamic reputations + rolling submodels) can inform follow-on work in federated personalization and model-heterogeneous CF at scale.

**Weaknesses:**

** Theoretical sssumptions**

> In Theorem 1 (*Fairness in training loss*), it assumes L-smoothness and µ-strong convexity, which is unusual and sometime opposite for deep networks. It also assumes that a higher-reputation client’s submodel contains another client’s submodel, which is not guaranteed by the actual allocation mechanism (submodels are selected by validation performance rank, not by inclusion nesting).

> Theorem 2 (*Convergence*) relies on Assumption 5 that “each neuron is assigned the same number of times after T rounds,” leading to a contraction model. This is a very strong structural assumption about learning dynamics and aggregation. My understanding is that it does not show that the proposed allocation actually enforces equal frequencies under client sampling and non‑IID data.

**Baseline choices and protocol**

> The paper does not include model-heterogeneous rolling baselines like FedRolex (which specifically addresses rolling extraction and neuron coverage) even though the proposed method borrows its rolling idea. This makes it hard to separate the effect of fairness-aware ranking from the benefit of rolling coverage.

> The appendix states: “we require all algorithms to allocate rewards according to client contributions,” i.e., re-implementing baselines under your reward protocol. That changes the original algorithms (CFFL, CGSV, etc.) and can be unfair. For example, those methods were designed with different reward pathways (e.g., gradient-Shapley-based sharing).

**Method specification and complexity/overhead**

> The submodel allocation uses quantity($r_i$,${A^t}$), i.e., mapping reputations to the set of validation scores of candidate submodels. It remains unclear how many submodels are evaluated per round, how they are ranked/assigned when reputations do not match a discrete set of accuracies, and what the runtime overhead is (server-side validation on all masks each round can be substantial).

> The dynamic reputation function depends on the hyperparameters including multi factors. Only one is ablated. The others may materially affect dynamics and fairness.

**Fairness metrics and settings could be more comprehensive**

> Pearson correlation can be high even if rewards differ by a global scale/shift. $\alpha$‑bounded CF is introduced but  the choices and practical ranges are not fully discussed (DIR(3.0/7.0) are relatively mild; consider lower α in Dirichlet splits).

> Since the method shares lineage with Shapley/CGSV ideas (valuing contributions), a qualitative comparison to those value-based perspectives could be instructive; some setups (adversarial or corrupted clients) are covered by ShapleyFL/robust FL and might expose different behavior of FedRAC.

**Scope limitations**

> Experiments use small MLPs for image tasks (CIFAR‑10 accuracies ~50%), which weakens claims about general utility. Prior work shows rolling helps large models, which would align with the motivation.

> Relationship to group fairness is orthogonal (your goal is CF), but a short contrast experiment with FairFed (group fairness) would contextualize the method’s trade-offs across fairness notions.

**Questions:**

Please consider my concerns in the weakness section. On top of that, please also answer following questions.

> Baseline protocols: You “require all algorithms to allocate rewards according to client contributions.” Could you re-run CFFL/CGSV/FedAVE/IAFL/FedSAC as originally specified (without re-allocating rewards) and report results under your fairness/accuracy/rate metrics? This will clarify how much of the gain stems from your reward protocol vs. the rolling mechanism.

> Comparison to FedRolex: Since your rolling mechanism resembles FedRolex, can you include a FedRolex baseline (with your fairness metric) to isolate the benefit of dynamic reputations and low-frequency prioritization?

> Complexity and scalability: How many submodels are evaluated per round and how large is the validation set? Please report server-side runtime and communication overhead compared to baselines, and discuss scalability to larger models / more clients.

> Hyperparameters: Provide ablation plots for $\alpha$ and $Con$, not only $\beta$. How robust are fairness/accuracy/rate to these choices? Does improper tuning risk over-helping low-contributing clients and harming CF later?

> Fairness regimes: Could you evaluate harsher non‑IID (e.g., Dirichlet $\alpha$=0.1 or 0.5) and report additional CF metrics (e.g., rank correlation, slope of reward vs. contribution)?

> Relation to fairness taxonomy: Please justify your method within the broader taxonomy (client fairness, collaborative fairness, group fairness) and briefly show how it interacts (positively or negatively) with group fairness on a dataset with sensitive attributes (using FairFed as a reference).

> Connection to FedAvg theory: How do your dynamics interact with known FedAvg non‑IID convergence issues? Can your dynamic reputations mitigate client drift in practice beyond the neuron-frequency heuristic?

**Details Of Ethics Concerns:**

The paper uses public datasets (CIFAR‑10, SVHN, EMNIST) and doesn't present any ethics concerns.

---

### Note · Authors · 2025-11-13

I have read and agree with the venue's withdrawal policy on behalf of myself and my co-authors.